# Metabolic Profiling and Transcriptional Analysis of Carotenoid Accumulation in a Red-Fleshed Mutant of Pummelo (*Citrus grandis*)

**DOI:** 10.3390/molecules27144595

**Published:** 2022-07-19

**Authors:** Congyi Zhu, Cheng Peng, Diyang Qiu, Jiwu Zeng

**Affiliations:** Key Laboratory of South Subtropical Fruit Biology and Genetic Resource Utilization (MOA) & Guangdong Province Key Laboratory of Tropical and Subtropical Fruit Tree Research, Institute of Fruit Tree Research, Guangdong Academy of Agricultural Sciences, Guangzhou 510640, China; zhucongyi@gdaas.cn (C.Z.); pengcheng@gdaas.cn (C.P.); qiudiyang@gdaas.cn (D.Q.)

**Keywords:** *Citrus grandis* ‘Tomentosa’, red-fleshed mutant, carotenoid accumulation, phytochemical compounds, volatile components, transcriptional analysis

## Abstract

*Citrus grandis* ‘Tomentosa’, commonly known as ‘Huajuhong’ pummelo (HJH), is used in traditional Chinese medicine and can *moisten* the lungs, resolve phlegm, and relieve coughs. A spontaneous bud mutant, named R-HJH, had a visually attractive phenotype with red albedo tissue and red juice sacs. In this study, the content and composition of carotenoids were investigated and compared between R-HJH and wild-type HJH using HPLC–MS analysis. The total carotenoids in the albedo tissue and juice sacs of R-HJH were 4.03- and 2.89-fold greater than those in HJH, respectively. The massive accumulation of carotenoids, including lycopene, β-carotene and *phytoene**,* led to the attractive red color of R-HJH. However, the contents of flavones, coumarins and most volatile components (mainly D-limonene and γ-terpinene) were clearly reduced in R-HJH compared with wild-type HJH. To identify the molecular basis of carotenoid accumulation in R-HJH, RNA-Seq transcriptome sequencing was performed. Among 3948 differentially expressed genes (DEGs), the increased upstream synthesis genes (phytoene synthase gene, PSY) and decreased downstream genes (β-carotene hydroxylase gene, CHYB and carotenoid cleavage dioxygenase gene, CCD7) might be the key factors that account for the high level of carotenoids in R-HJH. These results will be beneficial for determining the molecular mechanism of carotenoid accumulation and metabolism in pummelo.

## 1. Introduction

Carotenoids are a diverse group of colorful pigments that produce the yellow, orange and red colors in many fruits, vegetables and flowers [1]. Carotenoids and their derivatives have diverse biological functions in plant growth, development and reproduction [2]. They play important roles in the assembly of photosystems, in light harvesting, and in photoprotection [3]. Carotenoids are also important compounds for human nutrition and health due to their significant antioxidant function and serve as precursors for the biosynthesis of vitamin A [4,5].

As one of the most widespread fruit crops, citrus has the largest number of carotenoid species. More than 115 carotenoid compounds have been discovered in citrus, including β-carotene, lycopene, β-cryptoxanthin, zeaxanthin and neoxanthin [6,7]. Citrus varieties and bud mutations provide excellent materials for the analyses of carotenoid composition and carotenoid biosynthetic genes. The red-fleshed navel orange Cara Cara, a spontaneous mutant from the blonde-fleshed ′Washington Navel′, accumulated a high concentration of lycopene in the pulp, despite this carotene being absent in ordinary sweet oranges [8,9]. The other red-fleshed mutant Hong Anliu was discovered in China as a bud mutant derived from Anliu sweet orange. Hong Anliu orange also presented a 1000-fold higher carotenoid content than wild-type fruits [10]. *C. grandis* L. Osbeck (pummelo), with more than 200 cultivars, is widely cultivated in southern China, such as in Guangdong, Guangxi and Fujian provinces. It is considered the largest citrus fruit and is well known for its nutritional benefits [11,12]. *C. grandis* ‘Tomentosa’ (CGT) is a cultivar that originated in Huazhou town in Guangdong Province in southern China. Exocarpium *Citri Grandis* (ECG, Huajuhong in Chinese), the dried unripe fruit or fruit peel of CGT, is a well-known traditional Chinese medicine that is officially listed in the Chinese pharmacopoeia [13]. ECG has attracted increasing attention for its distinguished pharmacological heat-clearing, antitussive and expectoration effects [14,15]. Flavonoids, coumarins and limonoids are considered to be the main effective components [16,17].

Recently, we discovered a spontaneous bud mutant, named R-HJH, which had a visually attractive phenotype with red albedo tissue and red juice sacs (Figure 1). In this study, the composition and content of carotenoids were investigated and compared between R-HJH and wild-type HJH by using HPLC–MS analysis. The contents of three active ingredients, flavonoids, coumarins and volatiles, were also qualitatively measured by HPLC–DAD and GC–MS. To investigate the molecular mechanisms of carotenoid accumulation in R-HJH, we investigated the gene expression related to carotenoid biosynthesis and catabolism between R-HJH and wild-type HJH by using the RNA-Seq technique.

## 2. Results

### 2.1. Carotenoid Composition and Content

A total of 23 carotenoids were detected by liquid chromatography–mass spectrometry of the albedo tissue and juice sacs of R-HJH and HJH. As shown in Table 1, the quantity of total carotenoids in the albedo tissue of R-HJH was 15.553 ± 0.100 μg/g (dry weight), which was 4.03-fold greater than that in HJH. Lycopene in the albedo of R-HJH and wild-type HJH reached levels of 10.407 ± 0.315 μg/g and 0.2974 ± 0.189 μg/g, respectively, which accounted for 66.91% and 7.60% of the total carotenoids, respectively. β-carotene accumulated to 1.347 ± 0.096 μg/g in the albedo of R-HJH and was the second highest carotenoid, while it was 0.235 ± 0.035 μg/g in the albedo of HJH. The change trend of carotenoids in the juice sacs between R-HJH and wild-type HJH was consistent with that in the albedo. The total carotenoid content in the juice sacs of R-HJH was 2.89-fold greater than that in wild-type HJH (19.432 ± 1.104 μg/g vs. 6.720 ± 0.153 μg/g, *p* < 0.01). The quantity of lycopene in the juice sacs of wild-type HJH was 3.207 ± 0.061 μg/g, which accounted for 19.54% of the total carotenoids. However, the content of lycopene in the juice sacs of R-HJH was 11.333 ± 1.104 μg/g, which was 2.53-fold greater than that in HJH. The proportion of lycopene in the juice sacs of R-HJH reached 58.32%. The β-carotene level was significantly higher in the juice sacs of R-HJH than those of wild-type HJH (1.890 ± 0.191 μg/g vs. 0.200 ± 0.007 μg/g, *p* < 0.01). In addition, the phytoene contents in the albedo and juice sacs of R-HJH reached levels of 1.166 ± 0.259 μg/g and 3.017 ± 0.158 μg/g, respectively. However, phytoene was undetected in the albedo or the juice sacs of HJH.

### 2.2. Quantification of Flavones and Coumarins

In the present study, 2 flavones (naringin and rhoifolin) and 2 coumarins (bergapten and isoimperatorin) were identified in both R-HJH and wild-type HJH using UPLC-DAD. However, significant differences (*p* < 0.05) were observed in the contents of these four phytochemical compounds between R-HJH and wild-type HJH (Table 2). The contents of naringin and rhoifolin in the whole fruit of wild-type HJH reached 55.81 ± 2.48 mg/g and 10.37 ± 0.29 mg/g, respectively. For R-HJH, the contents of these two flavones in the whole fruit were reduced by more than 30.55% and 44.84%, respectively. The contents of the two coumarins were also significantly lower in R-HJH than in wild-type HJH. In relation to wild-type HJH, R-HJH showed a sharp decline in the content of isoimperatorin (63.16%). The average content of bergapten in R-HJH (0.24 ± 0.01 mg/g) was also lower than that in wild-type HJH (0.33 mg/g).

### 2.3. Identification of Volatile Compounds

GC–MS analysis is a popular and powerful technique to identify volatile compounds, and headspace solid-phase microextraction (HS-SPME) is more rapid, sensitive, efficient and solvent-free than conventional methods, including steam distillation and solvent extraction. Hence, a combined method of HS-SPME-GC-MS was carried out in this study to investigate the volatile components in different fruit tissues of R-HJH and wild-type HJH. The fruits were separated into two parts, i.e., flavedo and albedo. Appendix A shows the total ion chromatograms (TICs) of all samples. A total of 122 compounds that had more than 90% similarity with authentic standards of the NIST database were successfully identified in the volatile oil from the flavedo and albedo of R-HJH and wild-type HJH. Among these 122 volatile components, D-limonene was the most abundant volatile component in all samples, followed by β-myrcene, γ-terpinene and linalool.

The results showed that the contents of different volatile components varied among the different tissues of the two species (Table 3). In the flavedo samples, the content of most volatile components decreased significantly in R-HJH compared to wild-type HJH, with a 33.4–86.5% reduction. The most significantly decreased volatile component was γ-terpinene. The content of γ-terpinene in the flavedo of R-HJH was decreased to approximately one-eighth that in wild-type HJH (124.34 ± 18.61 μg/mg vs. 917.96 ± 78.94 μg/mg). The content of limonene, which was the predominant compound, was reduced to less than one-third that in wild-type HJH. However, the amount of dodecanal in the flavedo was not significantly different between R-HJH and wild-type HJH (88.59 ± 17.96 μg/mg vs. 91.55 ± 7.81 μg/mg). In terms of the albedo, the volatile components of R-HJH showed a decreasing trend, which was consistent with that in the flavedo. Approximately 94% of the volatile components notably declined in R-HJH compared to wild-type HJH, with a 32.1–87.1% reduction. The α-pinene and D-limonene contents in the albedo of R-HJH decreased to one half that of wild-type HJH. β-myrcene, α-muurolene and γ-muurolene were detected in the albedo of wild-type HJH but not in the albedo of R-HJH.

### 2.4. RNA-Seq Analysis

To further determine the molecular basis of the phenotypes observed in R-HJH, global gene expression profiling of R-HJH albedo compared with the wild-type HJH albedo was carried out. The number of raw reads ranged from 41,725,292 to 52,446,264 per cDNA library. A total of 41.30–51.95 million clean reads were obtained after stringent quality checks and data cleanup. The percentages of clean reads and Q20 reads were more than 98% and 97%, respectively. A total of 86.5–90.4% of the clean reads were mapped uniquely to the *C. grandis* reference genome, and only a small proportion of them (2.37–2.88%) mapped multiple regions on the genome (Appendix A). These data indicated that the read number and sequencing quality were sufficient for further analysis.

A total of 14,852 and 14,739 genes were expressed (average TPM ≥ 1) in R-HJH and wild-type HJH, respectively (Appendix A). Pearson’s correlation coefficients among the three biological replicates were high (R2 > 0.99). To identify differentially expressed genes (DEGs) between these two samples, DEG screening was conducted based on a threshold of two-fold change (FDR < 0.05). Compared to HJH, 3948 DEGs were identified in R-HJH, with 1668 upregulated and 2280 downregulated genes (Appendix A). The number of downregulated genes was higher than the number of upregulated genes.

To obtain the functional information for the DEGs and the related biological processes they participate in, GO and KEGG enrichment analyses were conducted. A total of 70.14% (2769) of the unigenes were identified and annotated to GO terms (Appendix A), and 37.03% (1462) of them were assigned to KEGG pathways (Appendix A). Based on their GO classifications, DEGs were grouped into three functional GO categories (Figure 2). The most significantly enriched GO terms in the biological process category were related to metabolic processes and cellular processes. For molecular function, catalytic activity was the most highly represented GO term, followed by binding. In the cellular component category, most DEGs were enriched in four categories: cell part, membrane part, membrane and organelle.

KEGG pathway enrichment analysis revealed that the most enriched pathway was the metabolic pathway (821 DEGs), followed by the genetic information processing (129 DEGs), environmental information processing (120 DEGs) and organismal systems (120 DEGs) (Figure 3). Only 39 DEGs were assigned to cellular processes. When we focused on carotenoid biosynthesis, 12 unigenes were annotated according to KEGG analysis, with seven upregulated and five downregulated.

### 2.5. DEGs Related to Carotenoid Accumulation

Twelve DEGs putatively involved in carotenoid biosynthesis and degradation were identified in the present study (Appendix A). The expression level of PSY, the first committed step, was significantly higher in R-HJH than in wild-type HJH (2.23-fold, *p* < 0.01). In contrast, the expression of CHYB, which converts orange β-carotene into yellow β-cryptoxanthin and zeaxanthin, was significantly decreased in R-HJH compared with wild-type HJH (0.49-fold, *p* < 0.01). In addition, carotenoid cleavage dioxygenases (CCDs) and 9-cis-epoxycarotenoid dioxygenase (NCED) play a crucial role in carotenoid degradation, which converts a wide range of carotenoids into colorless apocarotenoids. The expression of CCD7 was significantly decreased in R-HJH compared with wild-type HJH (0.46-fold, *p* < 0.01). However, the expression of NCED was increased (2.23-fold, *p* < 0.01). Herein, the expression balance between upstream synthesis genes (PSY) and downstream genes (CHYB and CCD7) might be the key biosynthetic genes accounting for the high level of carotenoids in R-HJH.

## 3. Discussion

Citrus fruits are ornamental and edible. Color is one of the main economic traits in citrus production. Red-fleshed citrus is more popular with planters and consumers than wild-type citrus due to its high economic and health value. In this study, we investigated the biochemical and molecular changes underlying the characteristic pigmentation of R-HJH. To the best of our knowledge, this is the first time that this kind of bud mutant has been reported in HJH.

Carotenoids are responsible for flesh color and nutritional quality in horticultural crops such as tomato and citrus. In this study, a comparative analysis of carotenoid content and composition in the juice sacs and albedo tissue of R-HJH and wild-type HJH was performed. Wild-type HJH contained a normal level of carotenoids, while the mutant R-HJH contained over three to four times the amount of carotenoids in the albedo tissue and juice sacs. Higher-than-normal accumulation of carotenoids in mutant citrus fruits has also been reported in the orange (*C. sinensis*) varieties Shara, Cara Cara and Hong Anliu, as well as the pummelo (*C. grandis*) varieties Huangjinmiyou and Hongroumiyou (red-fleshed pummelo) [8,9,10,18]. Based on data reported in these literature results, the total carotenoids in the juice sacs of R-HJH were lower than those in Huangjinmiyou and Hongroumiyou but still higher than those in the juice sacs of the red oranges Cara Cara and Hong Anliu. However, the color of the albedo tissue was red in R-HJH, while it was still white in red-fleshed pummelo, similar to normal pummelo. Little attention has been given to the nutritional value of the albedo tissue due to its nonedible nature in freshly consumed cultivars of pumelo. However, the whole fruits of wild-type HJH have been used in traditional Chinese herbal medicines [19]. The total carotenoid content of the albedo tissue in R-HJH was significantly higher than that in wild-type HJH, and there was a slight decrease in this content in the albedo tissue compared with the juice sacs in R-HJH. The same trend was observed for lycopene, β-carotene and γ-carotene. Phytoene, a colorless C40 carotenoid intermediate, was found to be enriched in the juice sacs and albedo of R-HJH. However, it was not present in the juice sacs or albedo of wild-type HJH. Phytoene has been reported to be the precursor of carotene and is converted into lycopene through successive desaturation reactions [20]. Based on these biochemical analyses, we found that the massive accumulation of linear carotenoids, such as lycopene and phytoene, the precursor of carotene, led to an attractive red color in the juice sacs and albedo of R-HJH.

Another interesting aspect of this study was the observation of the occurrence of a significant decrease in the flavone, coumarin and volatile contents in R-HJH. Plant secondary metabolites, such as carotenoids, volatile terpenoids, limonoids and ABA, belong to a large class of terpenoids and share the common precursor pyruvate. The content of these metabolites was closely related to the competitive ability of the relative metabolic pathways for their same precursor compounds. Similar results have been reported in other red-fleshed mutants, such as Cara Cara, red-Anliu oranges and red-fleshed Guanxi pummelo. Cara Cara and red-Anliu oranges showed increased contents of carotenoids and the limonoid aglycone and decreased contents of abscisic acid (ABA) catabolites, monoterpenoid volatiles, and sesquiterpenoid volatiles [21]. Liu et al. reported that the ABA contents in red-Anliu orange and red-fleshed Guanxi pummelo were significantly lower than those in their corresponding wild types [22].

With the development of molecular biotechnology, the carotenoid biosynthetic and metabolism pathways in citrus have been clarified, and some key genes have been cloned successfully. This study provides a global view of the gene expression changes in the red-fleshed mutant R-HJH compared to wild-type HJH. In R-HJH, the transcription of PSY, which catalyzes the production of the first carotenoid molecule phytoene, was higher than that in wild-type HJH. Several studies reported that the expression level of PSY was related to carotenoid accumulation in citrus [23,24]. Moreover, the expression of CHYB and CCD7, which play an important role in downstream lycopene biosynthesis, was upregulated in R-HJH compared to wild-type HJH. Previous studies demonstrated that the CHYB gene affected the color of citrus fruit by converting orange β-carotene into yellow β-cryptoxanthin and zeaxanthin through a hydroxylation reaction. It was reported that the CCD gene family was related to the cleavage of carotenoids into apocarotenoids at different double-bond positions [25,26]. Zhang et al. found that four carotenoid biosynthetic pathway genes, 1-deoxy-D-xylulose-5-phosphate synthase (DXS1), deoxyxylulose 5-phosphate reductoisomerase (DXR), geranylgeranyl diphosphate synthase (GGPPS2), and phytoene synthase (PSY1), were highly and positively related to lycopene levels in Cara Cara orange by a combination of metabolomic and transcriptomic analyses [27].Two upstream genes (PSY and ζ-carotene desaturase, ZDS) were upregulated and two downstream genes (LCYB and capsanthin–capsorubin synthase, CCS) were downregulated. These genes might be involved in the massive accumulation of lycopene in Hong Anliu sweet orange, consistent with the mechanism regulating lycopene accumulation in Cara Cara orange [28,29]. Similar results were reported in ‘Siam Red Ruby’, a new lycopene-accumulating pummelo variety in Thailand. Gene expression results showed that increases in the expression of upstream genes (CitPSY, CitPDS, CitZDS, CitZISO, and CitCRTISO) and decreases in the expression of downstream genes (CitLCYb1, CitLCYb2, CitLCYe, CitHYb and CitHYe) were the main mechanisms controlling lycopene accumulation in the pulp of ‘Siam Red Ruby’ [30].

## 4. Materials and methods

### 4.1. Plant Materials

The red-fleshed mutant R-HJH and the wild-type HJH pummelo cultivated in the production area in Ligang Town, Huazhou City, Guangdong Province, China, were used as plant materials. Fruit samples were collected from ten-year-old pummelo trees 150 days after flowering in 2020, and the flavedo tissue, albedo and juice vesicles were carefully separated with a scalpel. There were three biological replicates for each sample, and each biological replicate contained a mixture from ten representative fruits. The samples were then frozen immediately in liquid nitrogen and stored at −80 °C until analysis.

### 4.2. Carotenoid Extraction, Identification and Quantification

The carotenoids were extracted as described previously with some modifications [31,32]. Freeze-dried samples (50 mg) were ground into powder and extracted with 1.0 mL of a mixed reagent of n-hexane:acetone:ethanol (2:1:1, *V/V/V*) (containing 0.01% butylated hydroxytoluene, BHT). The extract was vortexed for 30 s, sonicated for 20 min and centrifuged at 12,000× *g* for 5 min. The extraction was repeated twice, and the supernatants were pooled, dried under a nitrogen gas stream and redissolved in methanol:methyl tert-butyl ether (3:1, *V*/*V*). Carotenoid quantification was analyzed using an LC-APCI-MS/MS system [UPLC, ExionLC™ AD (SHIMADZU, Kyoto, Japan); MS, AB Sciex 6500 Triple Quadrupole (Foster City, CA, USA)] in Metware (Wuhan Metware Biotechnology Co., Ltd., Wuhan 430070, China).

The chromatographic conditions were as follows: column, YMC C30 (3 μm, 2 mm × 100 mm); mobile phase A, acetonitrile:methanol (3:1, *V*/*V*) (0.01% BHT and 0.1% formic acid); mobile phase B, methyl tert-butyl ether (0.01% BHT); gradient program, 100:0 *V*/*V* at 0 min, 100:0 *V*/*V* at 3 min, 30:70 *V*/*V* at 5 min, 5:95 *V*/*V* at 9 min, 100:0 *V*/*V* at 10 min, and 100:0 *V*/*V* at 11 min; flow rate, 0.8 mL/min; temperature, 28 °C; and injection volume, 2 μL. After UPLC, the effluent was alternatively connected to an ESI-triple quadrupole-linear ion trap (Q TRAP)-MS.

MS/MS was performed with an Applied Biosystems 6500 Quadrupole Trap. The spectrometer was fitted with an atmospheric pressure chemical ionization (APCI) Turbo Ion-Spray interface operating in positive ion mode and controlled by Analyst 1.6.3 software (AB Sciex). The key parameters were as follows: ion source, APCI+; source temperature, 350 °C; curtain gas (CUR), 25.0 psi; and collision gas, medium. The declustering potential (DP) and collision energy (CE) for individual MRM transitions were obtained with further DP and CE optimization. A qualitative analysis was carried out according to the second-level spectral information, which is based on the self-established MWDB (the Metware database) and a public database of metabolite information. Metabolite quantification was accomplished by using multiple reaction monitoring (MRM) analysis.

### 4.3. Analysis of Phytochemical Compounds by UPLC-DAD

The samples were ground into fine powder using A100 with liquid nitrogen. A total of 0.8 g of the sample was extracted with 40 mL methanol by ultrasonication (30 min, 100 Hz) on ice. After centrifugation, the supernatant was filtered and stored at −20 °C before LC–MS analysis.

The analysis was performed on an Agilent 1290 series UPLC system (Agilent, Santa Clara, CA, USA) coupled with an Agilent G4212 UV detector. Samples were separated on an Agilent ZORBAX SB-C18 column (4.6 × 50 mm, 1.8 μm) at 25 °C. Liquid phase A, water (0.1% formic acid); Liquid phase B, acetonitrile (0.1% formic acid); the gradient program: 80:20 *v*/*v* at 0 min, 70:30 *v*/*v* at 15 min, 50:50 *v*/*v* at 25 min, 25:75 *v*/*v* at 35 min, 0:100 *v*/*v* at 40 min, 0:100 *v*/*v* at 50 min, and 80:20 *v*/*v* at 60 min; flow rate, 0.75 mL/min; temperature, 30 °C; and injection volume, 10 μL. The DAD was set to 283 nm for naringin, 310 nm for isoimperatorin, 320 nm for bergapten, and 330 nm for rhoifolin. The external standard method was used for quantitative identification. Naringin (Lot. No. 1226D021), rhoifolin (Lot. No. 727B021), isoimperatorin (Lot. No. 1229A025) and bergapten (Lot. No. 619B021) were purchased from Aladdin Reagent Co., Ltd., China. These standard samples were serially diluted with methanol to obtain the standard working solutions for quantification.

### 4.4. Volatile Organic Compounds Identified by HS-SPME and GC/MS

The samples were pretreated as described below: 0.2 g of pulverized sample was mixed with 8.0 mL Milli-Q water into a 20 mL headspace vial, and 200 μL of 1-pentanol was added as an internal standard (10 ppm). An SPME holder containing a fused-silica fiber coated with a 50/30 μm layer of divinyl-benzene/carboxen/polydimethylsiloxane (DVB/CAR/PDMS) was exposed in the head space of the vial for 20 min at 50 °C and then removed from the vial and introduced directly into the GC injector, where thermal desorption was performed at 250 °C for 5 min under 1 mL/min of gas flow. The analysis of the volatiles in the samples was performed using a GC–MS instrument (Agilent 7890B-5977A) equipped with an electron impact (EI) ion source. The volatiles were separated on an HP-5MS capillary column (30 m × 0.25 mm × 0.25 μm; Agilent, USA). The column oven temperature program was set as follows: 50 °C for 1 min, increasing to 145 °C at 5 °C/min, increasing to 175 °C at 7 °C/min, increasing to 195 °C at 5 °C/min, and then increasing to 250 °C at 3 °C/min and maintained at that temperature for 10 min. Helium (99.999%) was used as the carrier gas at a flow rate of 1.0 mL min^−1^. Mass spectra were recorded in the electron impact ionization mode at 70 eV, and the mass scanning range was set from 30 to 400 amu in full scan mode. The identification of volatile compounds was determined using the NIST Atomic Spectra Database version 1.6, their retention indices and mass spectra were combined with those reported in references [33]. The VOC content was expressed as mg/kg of sample fresh weight.

### 4.5. RNA Extraction, Library Construction, and RNA-Seq

Total RNA was extracted from samples using TRIzol Reagent according to the manufacturer’s instructions. RNA quality and quantity were then determined by a NanoDrop 2000 UV–vis spectrophotometer (Thermo Scientific, Waltham, MA, USA). RNA purification, library construction, and sequencing were completed by a professional transcriptome sequencing company (Majorbio, Shanghai, China). Briefly, a paired-end library was synthesized by using a TruSeq RNA Sample Preparation Kit (Illumina, San Diego, CA, USA) and then sequenced with a HiSeq X Ten sequencer(Illumina, San Diego, CA, USA) (2 × 150 bp read length).

### 4.6. Transcriptome Analysis

After quality control, the clean reads were mapped to the *Citrus grandis* genome (http://citrus.hzau.edu.cn/) (accessed on 30 October 2020) using HISAT2 software [34]. The expression levels of each transcript were calculated using RSEM software with the transcripts per million (TPM) method [35]. Differentially expressed genes between two samples were identified using DESeq2. DEGs with |log2FC| > 1 and Q value ≤ 0.05 were considered significantly differentially expressed [36]. In addition, the functional enrichment analysis of DEGs, including Gene Ontology (GO) and the Kyoto Encyclopedia of Genes and Genomes (KEGG) enrichment, was performed using Goatools and KOBAS software, respectively [37].

## 5. Conclusions

This is the first work to report the physiological and molecular characterization of carotenoid accumulation in R-HJH. The red pigmentation of the pummelo mutant R-HJH is due to the accumulation of lycopene, followed by β-carotene and *phytoene*, whereas these pigments were low or undetectable in wild-type HJH pummelo. Furthermore, our study provides a global overview of the transcriptomic profile of R-HJH. The increases in the expression of upstream synthesis genes (PSY) and the decreases in the expression of downstream genes (CHYB and CCD7) might be the key aspects that account for the lycopene accumulation in R-HJH. However, additional studies are required to confirm these findings.

## Figures and Tables

**Figure 1 molecules-27-04595-f001:**
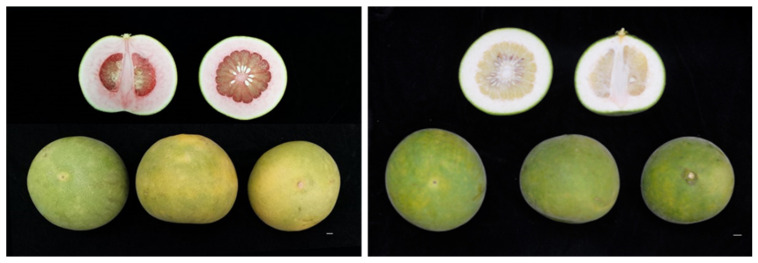
Internal appearance of red-fleshed pummelo (**left**) and wild-type HJH pummelo (**right**) fruit.

**Figure 2 molecules-27-04595-f002:**
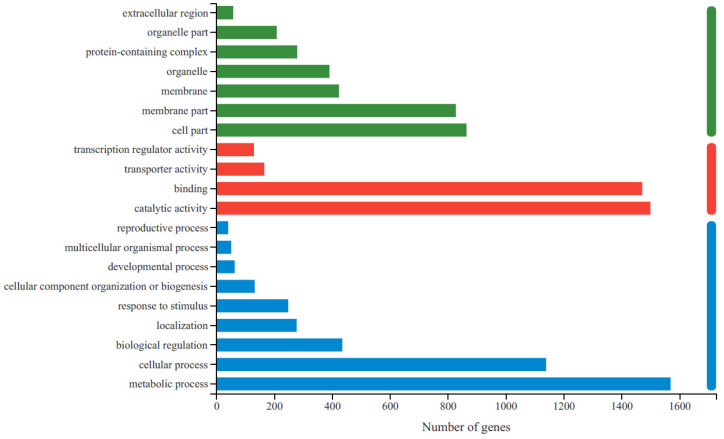
The statistics of GO functional classification of differentially expressed genes (DEGs) in R-HJH compared with HJH. The X-axis indicates the number of DEGs. The Y-axis represents GO terms. All GO terms are grouped into three ontologies: blue represents biological processes, red represents molecular functions and green represents cellular components.

**Figure 3 molecules-27-04595-f003:**
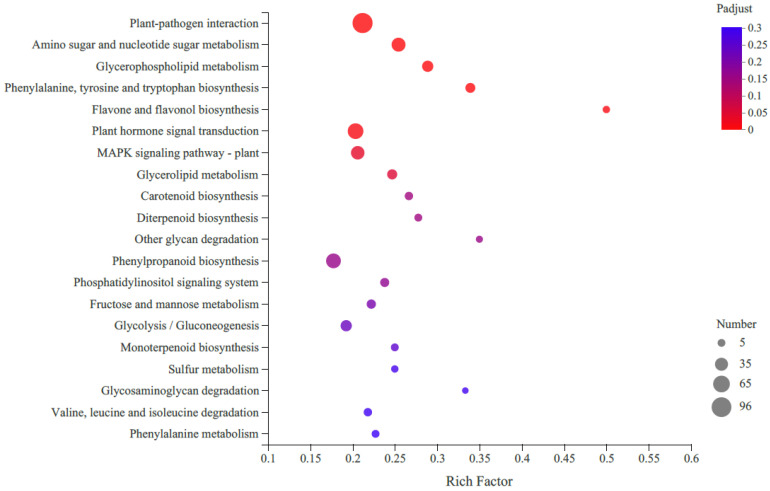
Top 20 enriched KEGG pathways of differentially expressed genes (DEGs) in R-HJH compared with HJH. The X-axis shows the rich factor. A high q-value is represented by blue, and a low q-value is represented by red (*p* value < 0.05). The Y-axis represents the second KEGG pathway terms. The number of DEGs is represented by the size of the circle.

**Table 1 molecules-27-04595-t001:** Carotenoids (μg/g DW) in the albedo and the juice sacs of R-HJH and their corresponding wild type HJH.

Sample	Total Carotenoids	Β-Carotene	Lutein	Zeaxanthin	Lycopene	Β-Cryptoxanthin	Γ-Carotene	Phytoene
R-HJH Albedo	15.553 ± 0.100	1.347 ± 0.096	1.193 ± 0.188	0.197 ± 0.017	10.407 ± 0.315	0.109 ± 0.031	0.268 ± 0.026	1.166 ± 0.259
HJH Albedo	3.864 ± 0.289	0.235 ± 0.035	2.593 ± 0.097	0.169 ± 0.016	0.294 ± 0.189	0.002 ± 0.001	0.076 ± 0.005	-
R-HJH juice sacs	19.432 ± 1.104	1.890 ± 0.191	1.600 ± 0.047	0.689 ± 0.084	11.333 ± 0.636	0.075 ± 0.008	0.341 ± 0.024	3.017 ± 0.158
HJH juice sacs	6.720 ± 0.153	0.200 ± 0.007	1.667 ± 0.058	1.313 ± 0.030	3.207 ± 0.061	0.003 ± 0.001	0.110 ± 0.003	-

**Table 2 molecules-27-04595-t002:** The content of flavone and coumarin (mg/g) in the whole fruit of R-HJH and HJH.

	HJH	R-HJH
naringin	55.81 ± 2.48	38.76 ± 1.82
rhoifolin	10.37 ± 0.29	5.72 ± 0.15
isoimperatorin	0.95 ± 0.01	0.35 ± 0.01
bergapten	0.33	0.24 ± 0.01

**Table 3 molecules-27-04595-t003:** Main volatiles compounds (µg/mg) in two different fruits tissues of R-HJH and HJH.

No	Name	CAS#	RI	Exocarp		Pulp	
		No.		HJU	R-HJH	HJU	R-HJH
1	α-Pinene	80-56-8	7.103	-	-	5.58 ± 0.29	2.53 ± 0.18
2	β-Myrcene	123-35-3	8.613	813.58 ± 74.28	268.86 ± 42.92	169.82 ± 7.53	
3	α-Phellandrene	99-83-2	8.993	173.41 ± 14.46	78.72 ± 8.32	11.28 ± 1.26	6.59 ± 0.63
4	D-Limonene	5989-27-5	9.757	8594.44 ± 700.36	5721.00 ± 827.98	774.94 ± 43.00	398.28 ± 22.36
5	γ-Terpinene	99-85-4	10.537	917.96 ± 78.94	124.34 ± 18.61	103.85 ± 4.20	42.47 ± 2.61
6	Linalool	78-70-6	11.679	1189.20 ± 26.86	915.71 ± 22.37	76.03 ± 2.76	37.68 ± 0.65
7	Nonanal	124-19-6	11.791	454.39 ± 371.06	63.48 ± 8.13	6.74 ± 0.66	4.58 ± 0.38
8	α-Terpineol	10482-56-1	14.302	467.10 ± 8.22	231.96 ± 4.00	26.97 ± 1.24	16.59 ± 0.55
9	Neral	106-26-3	15.676	722.94 ± 17.79	430.55 ± 18.60	18.52 ± 0.68	9.59 ± 0.29
10	Geraniol	106-24-1	16.017	95.81 ± 1.86	63.39 ± 1.40	6.76 ± 0.30	7.18 ± 0.47
11	Citral	5392-40-5	16.485	1036.55 ± 24.90	620.00 ± 26.35	29.05 ± 1.41	18.27 ± 0.58
12	Dodecanal	112-54-9	20.102	91.55 ± 7.81	88.59 ± 17.96	4.73 ± 0.48	1.72 ± 0.03
13	Caryophyllene	87-44-5	20.532	324.47 ± 33.88	134.35 ± 36.49	12.14 ± 1.31	2.16 ± 0.21
14	γ-Muurolene	30021-74-0	21.925	223.37 ± 25.60	127.79 ± 33.25	9.73 ± 1.07	-
15	Germacrene D	23986-74-5	22.070	1085.83 ± 110.57	591.64 ± 182.15	32.36 ± 3.29	4.17 ± 0.37
16	α-Muurolene	10208-80-7	23.372	111.13 ± 10.70	65.07 ± 15.59	4.31 ± 0.46	-

“#” indicated Chemical Abstracts Service. “-” indicated not detected.

## Data Availability

Not applicable.

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
