# Peer review of "Metabolic Profiling and Transcriptional Analysis of Carotenoid Accumulation in a Red-Fleshed Mutant of Pummelo (Citrus grandis)"

_molecules, 2022, doi:10.3390/molecules27144595_

Round 1

Reviewer 1 Report

The article of Zhu et al. “Metabolic Profiling and Transcriptional Analysis of Carotenoid Accumulation in a Red-Fleshed Mutant of Pummelo (Citrus grandis)” describes identification of carotenoids in red-variant of pummelo. The correlation of expression of upstream and downstream genes is also provided. The authors used appropriate methods for analysis and research is well designed with exception of detection of volatile organic compounds. The authors used pulverized sample (which is obtained by freeze-drying), thus the content of volatile compounds can be significantly altered by sample preparation method. Generally, the article is well written and its publication will be significant for the field.

There are some minor formatting issues:

Table 1, please, unify the labels of columns i.e. either start them with capital or not but keep it constant.

The charts in supplementary file – Fig. S1 – should have labeled axis.

Author Response

There are some minor formatting issues:

Comment1. Table 1, please, unify the labels of columns i.e. either start them with capital or not but keep it constant.

We have revised as requested.

Comment 2. The charts in supplementary file – Fig. S1 – should have labeled axis.

We have revised as requested.

Reviewer 2 Report

Dear respected authors,

I carefully checked this paper. This is an interesting manuscript dealing with phytochemical and molecular assays. Please note that all manuscripts submitted to Molecules should be critically investigated to improve their content. However, before publishing amending some comments is recommended. In this regard, the following comments should be considered before the publication of this paper.

1-     Lines 31-33: please add a reference.

2-     Overall, introduction is too long, please summarizing this section into three paragraphs. The rest of the introduction can move to the discussion section.

3-     Please add the whole-plant figure of both studied cultivars.  

4-      Lines 70-72: please add a reference.

5-     Figure 1: if this figure was previously published in your recent work as discussed in the text, please provide its reference for academic readers.

6-     Please correct figure 1 caption. The inserted detail is not clear.

7-     Please provide a flowchart for the M&M section.

8-     Section 4.1: please provide supplementary images for the cultivation of the studied cultivars

9-     Lines 272-298: please mention which references or previously published protocols were used for preparing your extracts.

10- Chromatograms in supplementary files have low quality. Please increase the resolution of these graphs and try to show the identified compounds based on the observed peaks in chromatograms.

11- Please supplement the results of RNAseq annotation in supplementary files. Table 1 in supplementary files only represents the numerical values of the final analyses.

12- Please determine which genes or transcription factors are up/down-regulated in the studied cultivar(s)

13- Why did the respected authors upload two similar supplementary files? If both cultivars have been used for transcriptomic assays, please supplement the details of each cultivar separately.

14- GO analyses were not conducted correctly. Please use at least two tools to highlight the most GO terms for analyzed transcriptomes.

15- Figure 3: Please supplement the rest of the enriched KEGG pathways. In this figure, only the top 20 enriched pathways have been shown. Please also use Pathvisio (https://pathvisio.org/) and Cytoscape (preferably Cluego module: https://apps.cytoscape.org/apps/cluego) to conduct GO analysis using identified gene sequences.

16- Please provide the chemical structure of identified compounds. If there were special stereochemistry for identified compounds, please add it to the structure.

17-  The authors mentioned that “In the present study, 2 flavones (naringin and rhoifolin) and 2 coumarins (bergapten 102 and isoimperatorin) were identified in both R-HJH and wild-type HJH using UPLC-DAD” would you please kindly determine which gene or genes are involved in the biosynthesis of these compounds according to RNAseq data?

18- Please clearly address all up-regulated genes in association with top identified compounds.

19- The analysis of RNAseq data requires further analysis to show the variation of expressed genes linked to identified compounds.

20- Please map the RNAseq data with KEGG pathways for flavonoids and coumarin synthesis routes. This analysis returns a number of expressed genes associated with identified compounds.

21- Authors used a simple RNAseq analysis to study the observed variations between the studied phenotypes. Please clearly mention how RNAseq does this because in such cases the cultivated plant should be undergone a specific situation to show the desired phenotype.

22- The respected authors mentioned that: “A total of 14,852 and 14,739 genes were expressed (average TPM ≥ 1) in R-HJH and 153 wild-types HJH” please supplement the annotation files of these genes in additional files linked to the article. Please compare the datasets together using Venndiagram, preferably using the OrthoVenn2 tool (https://orthovenn2.bioinfotoolkits.net/), to highlight commonly expressed genes in both sets and unique genes for each cultivar.

23- Lines 184-197: please determine which genes are differentially expressed.

Overall, the manuscript has a good structure. However, the RNAseq section should be re-analyzed to provide more clear authors for academic readers. By interpreting the current data represented for this section, the reader cannot know which genes are critical in the biosynthesis of the identified compounds. In this regard, the article has serious flaws, and additional experiments are needed, and research is not conducted correctly

Author Response

  • Lines 31-33: please add a reference.

We have added two references (ref. 1-2).

  • Overall, introduction is too long, please summarizing this section into three paragraphs. The rest of the introduction can move to the discussion section.

We have simplified the introduction into three paragraphs, and removed “the molecular mechanism of carotenoid accumulation” to the discussion section.

  • Please add the whole-plant figure of both studied cultivars.  

There is no difference in the morphological characteristics of tree or leaf between R-HJH and the wild-type HJH. Therefore, the internal appearance of fruit was shown in figure 1.

  • Lines 70-72: please add a reference. 59-61

The content in Lines 70-72 is about the experiment content of our paper. No reference are required.

  • Figure 1: if this figure was previously published in your recent work as discussed in the text, please provide its reference for academic readers.

Figure 1 has not been used in any publishment.

  • Please correct figure 1 caption. The inserted detail is not clear.

There was a mistake about figure 1 caption. We have revised as requested.

  • Please provide a flowchart for the M&M section.

The content of this manuscript was followed the subtitle in M&M section. And the flowchart was not required for M&M section.

8- Section 4.1: please provide supplementary images for the cultivation of the studied cultivars

There is no difference in the morphological characteristics of tree or leaf between R-HJH and the wild-type HJH. And the tree of R-HJH and the wild-type HJH cultivated in the same orchard under the regular field management measures.

9- Lines 272-298: please mention which references or previously published protocols were used for preparing your extracts.

We have added two references (ref. 30-31).

10- Chromatograms in supplementary files have low quality. Please increase the resolution of these graphs and try to show the identified compounds based on the observed peaks in chromatograms.

We have corrected it. Due to the format conversion, the resolution of GC/MS total ion chromatograms (Fig S1) significantly decreased.

11- Please supplement the results of RNAseq annotation in supplementary files. Table 1 in supplementary files only represents the numerical values of the final analyses.

12- Please determine which genes or transcription factors are up/down-regulated in the studied cultivar(s)

Response to comments 11 and 12, DEGs including TFs were shown in Table S2 with GO, KEGG, NR, Swiss-Prot and Pfam annotion.

13- Why did the respected authors upload two similar supplementary files? If both cultivars have been used for transcriptomic assays, please supplement the details of each cultivar separately.

We have corrected it.

14- GO analyses were not conducted correctly. Please use at least two tools to highlight the most GO terms for analyzed transcriptomes.

GO functional classification of differentially expressed genes (DEGs) was shown in Table S3. The library construction, sequencing and analysis were completed by a professional transcriptome sequencing company (Majorbio, Shanghai, China). Goatools software was used for GO term enrichment analysis in Majorbio cloud platform ( https://cloud.majorbio.com/) following some published papers, such as “Shi et al. Nanohole-boosted electron transport between nanomaterials and bacteria as a concept for nano-bio interactions. Nat Commun. 2021 Jan 21;12(1):493. doi: 10.1038/s41467-020-20547-9.”.

15- Figure 3: Please supplement the rest of the enriched KEGG pathways. In this figure, only the top 20 enriched pathways have been shown. Please also use Pathvisio (https://pathvisio.org/) and Cytoscape (preferably Cluego module: https://apps.cytoscape.org/apps/cluego) to conduct GO analysis using identified gene sequences.

The KEGG enrichment analysis of differentially expressed genes (DEGs) in R-HJH compared with wild-type HJH was shown in Table S4.

16- Please provide the chemical structure of identified compounds. If there were special stereochemistry for identified compounds, please add it to the structure.

As we known, there is no stereochemistry for naringin, rhoifolin, bergapten and isoimperatorin. Chromatographic assay is often used as a general method for the detection of carotenoid (LC-MS/MS), flavones and coumarins (HPLC-DAD), and volatiles (GC-MS) based on the retention time, UV absorption spectrum and m/z value, and compared with data of reference standard and public database.

17- The authors mentioned that “In the present study, 2 flavones (naringin and rhoifolin) and 2 coumarins (bergapten 102 and isoimperatorin) were identified in both R-HJH and wild-type HJH using UPLC-DAD” would you please kindly determine which gene or genes are involved in the biosynthesis of these compounds according to RNAseq data?

18- Please clearly address all up-regulated genes in association with top identified compounds.

19- The analysis of RNAseq data requires further analysis to show the variation of expressed genes linked to identified compounds.

20- Please map the RNAseq data with KEGG pathways for flavonoids and coumarin synthesis routes. This analysis returns a number of expressed genes associated with identified compounds.

Response to comments 17-20, this manuscript compared the content of naringin, rhoifolin, bergapten, isoimperatorin and volatiles in R-HJH and wild-type HJH, due to these components are considered as the main effective components of CGT which is used in the Chinese pharmacopoeia. Most compounds are the end products of secondary metabolite pathway, and the standards of some intermediate products are difficult to obtain. Due to the absence of content of intermediate product, the DEGs involved in the secondary metabolite pathway was not analyzed in this manuscript.

21- Authors used a simple RNAseq analysis to study the observed variations between the studied phenotypes. Please clearly mention how RNAseq does this because in such cases the cultivated plant should be undergone a specific situation to show the desired phenotype.

The red-fleshed mutant R-HJH was a spontaneous bud mutant of Citrus grandis ‘Tomentosa’. The phenotype with red albedo tissue and red juice sacs in R-HJH has no relationship with cultivation conditions. The tree of R-HJH and the wild-type HJH cultivated in the same orchard under the regular field management measures.

22- The respected authors mentioned that: “A total of 14,852 and 14,739 genes were expressed (average TPM ≥ 1) in R-HJH and 153 wild-types HJH” please supplement the annotation files of these genes in additional files linked to the article. Please compare the datasets together using Venndiagram, preferably using the OrthoVenn2 tool (https://orthovenn2.bioinfotoolkits.net/), to highlight commonly expressed genes in both sets and unique genes for each cultivar.

Compared with the DEGs, commonly expressed genes cannot explain the changed phenotype in molecular level. Hence, commonly expressed genes did not supply in this manuscript. And the DEGs were shown in Table S2.

23- Lines 184-197: please determine which genes are differentially expressed.

DEGs involved in carotenoid accumulation were shown in Table S5.

Round 2

Reviewer 2 Report

I have no further comments on this paper. The respected authors answered all queries. 

Author Response

We are grateful for the referees’ suggestion. Those suggestions are valuable and very helpful for revising and improving our manuscript. We have revised the manuscript accordingly.